# Genomic epidemiology of the early stages of the SARS-CoV-2 outbreak in Russia

Andrey B. Komissarov [1,7], Ksenia R. Safina [2,3,7], Sofya K. Garushyants[3,7], Artem V. Fadeev [1],
Mariia V. Sergeeva[1], Anna A. Ivanova [1], Daria M. Danilenko [1], Dmitry Lioznov[1,4], Olga V. Shneider[5],
Nikita Shvyrev[6], Vadim Spirin[6], Dmitry Glyzin[6], Vladimir Shchur[6] & Georgii A. Bazykin [2,3✉]

The ongoing pandemic of SARS-CoV-2 presents novel challenges and opportunities for the use of phylogenetics to understand and control its spread. Here, we analyze the emergence of SARS-CoV-2 in Russia in March and April 2020. Combining phylogeographic analysis with travel history data, we estimate that the sampled viral diversity has originated from at least 67 closely timed introductions into Russia, mostly in late February to early March. All but one of these introductions were not from China, suggesting that border closure with China has helped delay establishment of SARS-CoV-2 in Russia. These introductions resulted in at least 9 distinct Russian lineages corresponding to domestic transmission. A notable transmission cluster corresponded to a nosocomial outbreak at the Vreden hospital in Saint Petersburg; phylodynamic analysis of this cluster reveals multiple (2-3) introductions each giving rise to a large number of cases, with a high initial effective reproduction number of 3.0 [1.9, 4.3].

[1] Smorodintsev Research Institute of Influenza, Saint Petersburg, Russia. [2] Skolkovo Institute of Science and Technology (Skoltech), Moscow, Russia. [3] A.A. Kharkevich Institute for Information Transmission Problems of the Russian Academy of Sciences, Moscow, Russia. [4] First Pavlov State Medical University, Saint Petersburg, Russia. [5] Vreden Russian Research Institute of Traumatology and Orthopaedics, Saint Petersburg, Russia. [6] National Research University Higher School of Economics, Moscow, Russia. [7] These authors contributed equally: Andrey B. Komissarov, Ksenia R. Safina, Sofya K. Garushyants. ✉email: g.bazykin@skoltech.ru

Since May 11, 2020, Russia is among the four countries with the highest number of confirmed COVID-19 cases[1]. However, the outbreak in Russia started later than in many neighboring European countries[2–4], possibly in part due to early implementation of non-pharmaceutical interventions (NPIs) limiting virus import. Early NPIs included introduction of quarantine for passengers arriving from China on January 23, closing the land border with China on January 31, cancellation of most incoming flights from China on February 1, restricting entrance of non-Russian citizens from China on February 4, and restricting entrance from Iran and South Korea in late February[5–13]. While the earliest formally confirmed two cases in Russia dated to January and could be associated with direct introduction from China[14], no further cases were detected until March 2, 2020, when a man returning from Italy tested positive[15].

Nevertheless, since March 3, a steady increase in confirmed cases has started, with the initial country-wide estimated reproduction number $R_t$ of ~2[16]. Before March 21, all confirmed Russian cases were imported, while most European countries already had local transmission by this time[2,17]. Since early March, Russian regional authorities had been implementing their own NPIs. In particular, specific measures were introduced in Moscow and Saint Petersburg, the two largest transportation hubs responsible, respectively, for 67%[18–20] and 10%[21] of Russia's international air traffic. In Moscow, since March 5, all international travelers were temperature checked at the border; and passengers coming from countries with registered cases of SARS-CoV-2 had to report to authorities; while those coming from countries with high case counts at the time, including China, Italy, Spain and the UK, were quarantined[22]. Since March 14, mandatory quarantine was also applied to passengers' family members[23], and since March 16, it was introduced for all international travelers[24]. The NPIs at Saint Petersburg were timed similarly[25]. On March 13, the entrance of non-Russian citizens from Italy was restricted at the state level[26], and on March 18, entrance into Russia for all non-Russian citizens[26] for non-emergency reasons was banned[27]. While inbound flights, mainly returning Russian citizens from abroad, were still operating as of early July, passenger traffic has decreased ~100-fold[28].

Here, we report an analysis of 211 SARS-CoV-2 complete genome sequences obtained in Russia between March 11 (when there were just 28 confirmed cases Russia-wide) and April 23 (when there were 62773 confirmed cases)[29,30]. Phylogenetic analysis reveals distinct introduced lineages associated with transmission within Russia, as well as multiple individual samples phylogenetically intertwined with non-Russian sequences. The largest identified lineage corresponds to an outbreak at the Vreden hospital; phylodynamics analysis of this outbreak reveals between 2 and 3 distinct introductions and initial rapid spread curbed by subsequent establishment of quarantine.

## Results

**Sampling and data acquisition.** Samples were obtained from hospitals and out-patient clinics as part of COVID-19 surveillance and sequenced at the Smorodintsev Research Institute of Influenza. We sequenced complete genomes of 135 samples from Russia, including 133 from Saint Petersburg, 1 from the Leningrad region, and 1 from the Republic of Buryatia. Samples were obtained between March 15 and April 23. For analysis, we combined this dataset with additional 76 genomes from Russia available at GISAID[31] as of May 26, 2020, obtained between March 11 and April 14. The resulting dataset includes 211 sequences from 25 out of the 85 regions (federal subjects) of Russia (including the Republic of Crimea), with the two regions with the largest numbers of cases, Moscow and Saint Petersburg,

most densely covered. Therefore, while coverage differs between regions, this dataset is representative of the early outbreak in Russia in terms of geographic spread (Fig. 1a). For phylogenetic context, we also used the 19,623 whole-length, high-quality GISAID genomes from the rest of the world available on May 26, 2020.

**Multiple origins of SARS-CoV-2 in Russia.** Phylogenetic analysis indicates that the Russian samples are scattered across the SARS-CoV-2 evolutionary tree, representing much of its global diversity. Most samples correspond to the B.1, B.1.1, and B.1.* lineages (PANGOLIN nomenclature[32]) or clade G, GR, and GH (GISAID nomenclature[33]) which are wide-spread in Europe (Figs. 2–3). While the predominantly Asian A, B, and B.2 lineages comprised 53% of the sampled global viral diversity around the time of Russian border closure (March 27), only 4 (2%) of the Russian samples belonged to them.

We aimed to identify distinct introductions of SARS-CoV-2 into Russia. Phylogenetically, each of the 211 Russian sequences belongs to one of the three categories (Fig. 4). Firstly, 77 (36%) of these sequences form the 9 distinct Russian transmission lineages (Figs. 2–4), defined as monophyletic groups (clades) carrying more than one sequence all of which are Russian. These lineages indicate within-Russia transmission of introduced variants. Three of these lineages had no Russian sequences at their ancestral nodes, indicating that they originated from at least three distinct introduction events (Fig. 5c, d; Supplementary Note).

The remaining six Russian transmission lineages carried both non-Russian and Russian sequences at their ancestral nodes (Fig. 5a, b). Such lineages, hereafter referred to as "stem-derived transmission lineages", could also result from distinct introduction events; alternatively, their last common ancestor could already reside in Russia. To estimate the number of introductions giving rise to the stem-derived lineages, we make use of the direct data on travel history (or lack thereof) available for a fraction of our patients. Using a statistical model, we estimate that these lineages together resulted from roughly three additional introduction events (Supplementary Note). This number could be an underestimate due to undersampling of diversity outside Russia. Indeed, one of the identified lineages (Fig. 5b) involves two samples that had travel history to two different countries, indicating likely double introduction within the same lineage.

Secondly, we observe 73 (34%) singletons that are not involved in any of the Russian transmission lineages, each possessing their own characteristic mutations not shared by any other Russian sequences (Fig. 6). These include 33 singletons without any Russian ancestral sequences, and 40 singletons stemming from ancestral nodes with Russian sequences (hereafter, "stem-derived singletons"). We assume that the former correspond to sole introduced cases, for a total of 33 such introductions. Most of them had probably not resulted in any within-Russia transmission. However, we find that some of the singleton sequences were sampled from patients without any travel history (Fig. 6c). This indicated that at least some of the singletons likely correspond to distinct introductions that yielded domestic transmission clusters, of which just one representative was sequenced. Using travel data, we estimate that stem-derived singletons resulted from ~6 additional introduction events (Supplementary Note).

Thirdly, the remaining 61 sequences (29%) fell into 12 sets of two or more identical sequences, each of which was also identical to some of the non-Russian sequences (Fig. 6d). These sets are further referred to as stem clusters. Again, individual samples within a stem cluster could correspond to distinct introductions or domestic transmission. When data on travel history is available, we find that some of such clusters include multiple individuals

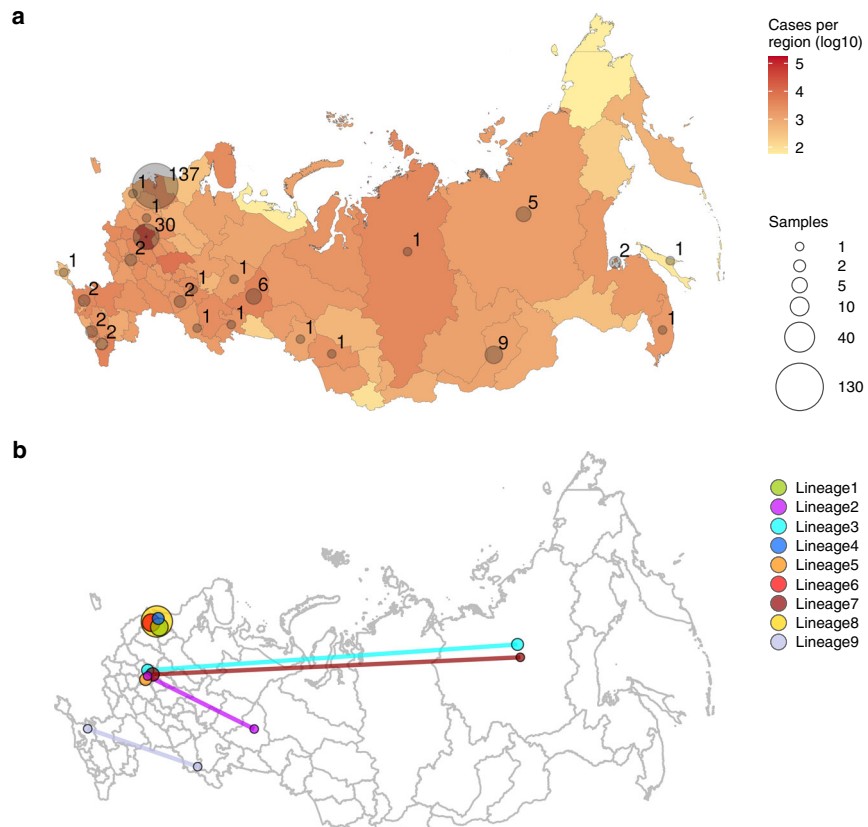

**Fig. 1 Early epidemiology of SARS-CoV-2 in the Russian Federation. a** The number of confirmed cases (**a**, red) and number of sequenced complete genomes (**a**, gray circles) per region of the Russian Federation (including the Republic of Crimea) as of May 26. Circle sizes are proportional to the numbers of obtained sequences, which are also shown next to the circles. For the purpose of this figure, Moscow was pooled with the surrounding Moscow Region, and Saint Petersburg was pooled with the surrounding Leningrad Region. **b** Identified Russian lineages and corresponding regions; circle size is proportional to the number of sequences belonging to the lineage at this region, and lineages spanning multiple regions are connected by lines. The initial map of Russian regions was downloaded from GADM[76].

with travel history, suggesting that identical sequences were repeatedly introduced into Russia at least in some instances (Fig. 6d). On the other hand, we also observe individuals without travel history, indicating domestic transmission of these variants. From travel data, the estimated number of introductions leading to stem cluster sequences is ~22.

Overall, we estimate the number of independent transmissions into Russia as ~6 resulting in transmission lineages, ~39 resulting in singletons, and ~22 resulting in stem clusters, for a total of 67 events. The uncertainty associated with this estimate is largely dependent on the approach for treating the numbers of introductions leading to stem clusters and stem-derived single-tons. If each stem cluster (together with any singletons derived from) is assumed to originate from exactly one introduction, the estimated number of introductions is 48. If instead each sequence within a stem cluster and each stem-derived singleton has resulted from a distinct introduction, the estimated number of introductions rises to 143.

The earliest collection date of a sample belonging to a transmission lineage represents the latest possible date this lineage could have been introduced into Russia. For most Russian transmission lineages, the earliest sample collection dates fall into the range between March 11 and 24, indicating that the corresponding lineages were introduced not long before (Fig. 7b). Indeed, out of the nine Russian transmission lineages, only two (lineages 6 and 8) had later dates of the earliest sequences. However, those were stem-derived lineages, and the oldest stem sequences corresponding to them dated to March 13, suggesting

that these transmission lineages could have also been established by this date. Many (15 out of 33) of the singletons were also collected within this timeframe, although some were collected later (mean date: March 29); together with the fact that many of the singletons have not traveled (Fig. 6, Supplementary Figs. 2–3), this indicates that they in fact correspond to as yet unsampled transmission lineages. By contrast, most stem-derived singletons were sampled at later dates (mean date: April 7, Mann–Whitney U-test, $p = 0.014$), suggesting that they were more likely than non-stem-derived singletons to originate from within-Russian transmission.

By the time introduction into Russia had started, the virus had already spread through other countries, with the same variant frequently present at multiple locations. Therefore, the source of most introductions could not be established unambiguously. Still, for a fraction of the samples, phylogenetic position is consistent with the source. For example, the earliest patient with known travel history has returned to Russia from France, and her sample is nested within a clade with just French and Danish sequences at the ancestral node, with French having earlier dates and therefore arguably more plausible source (Fig. 6a). For two additional sequences corresponding to regional outbreaks, no direct travel data was available but the probable source could be established from media reports, and was consistent with the phylogenetic position of the corresponding clades. This was the case for the import of clades from Switzerland into Yakutia (the Sakha Republic) (Fig. 5c)[34] and from Saudi Arabia to the Chechen Republic (Fig. 6b)[35].

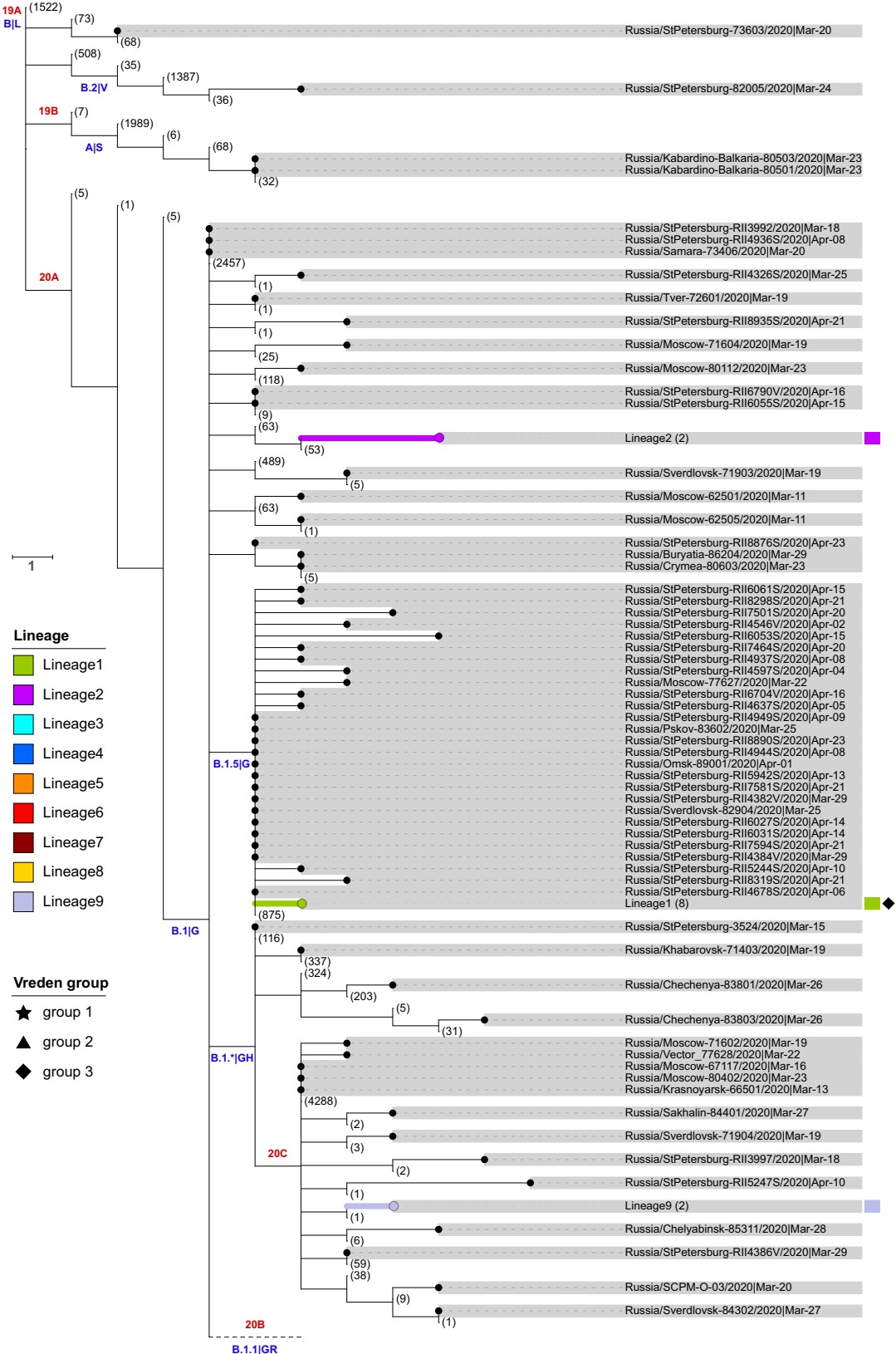

**Fig. 2 Phylogeny of SARS-CoV-2 in Russia.** Russian sequences are identified with dots and highlighted in gray. Russian transmission lineages are truncated to the founder node and highlighted with color (the color scheme is consistent between Figs. 1–2 and 4–6). Major SARS-CoV-2 lineages are labeled according to Nextstrain[81] and PANGOLIN|GISAID nomenclature in red and blue, respectively. Non-Russian sequences and lineages carrying no Russian sequences are truncated, with numbers of such sequences shown in brackets. Sequences from the Vreden hospital and lineages carrying such sequences are marked with star, triangle and diamonds. Branch lengths represent the number of nucleotide substitutions. "hCoV-19/" prefixes are excluded from all sample names for clarity. For readability, the B.1.1 lineage is collapsed (dashed line) and shown separately in Fig. 3.

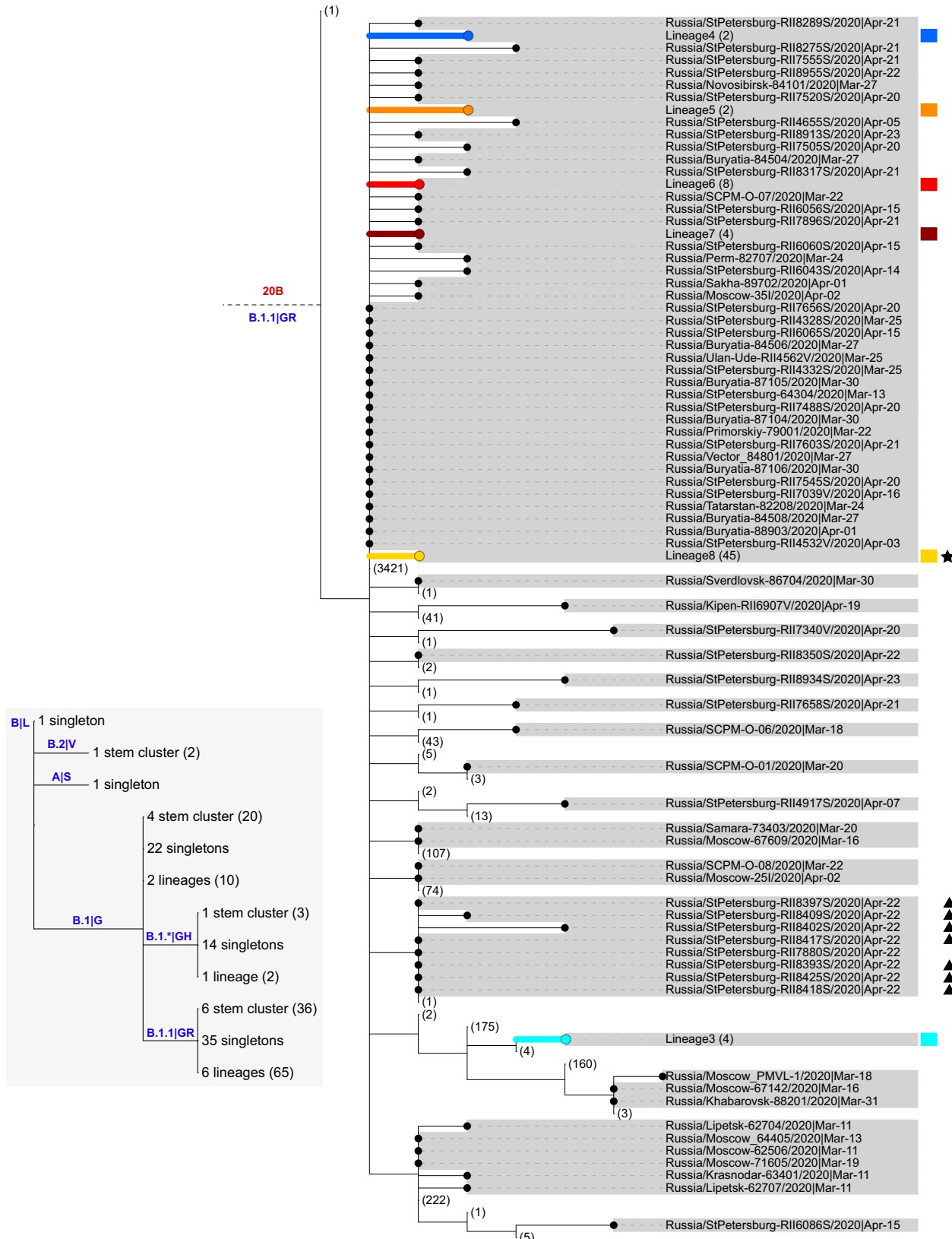

**Fig. 3 Phylogeny of SARS-CoV-2 in Russia, lineage B.1.1.** Notation is the same as in Fig. 2. The inset summarizes the distribution of Russian singletons, stem clusters and transmission lineages across major SARS-CoV-2 clades.

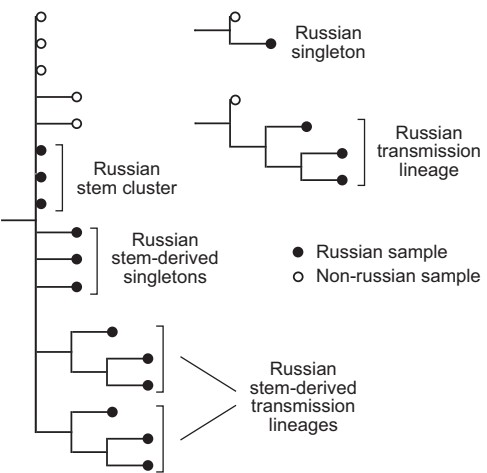

**Fig. 4 Terminology for phylogenetic groups of samples.** We categorized Russian samples into five categories: Russian transmission lineage-set of two or more sequences that form a Russian-only clade; Russian singleton-single Russian sequence that forms a clade of its own and is not a part of a Russian transmission lineage; Russian stem cluster-set of Russian sequences identical to each other and to some non-Russian sequences; Russian stem-derived transmission lineage-a Russian transmission lineage whose immediate ancestor is a Russian stem cluster; and Russian stem-derived singleton-Russian singleton whose immediate ancestor is a Russian stem cluster.

Overall, out of the 13 patients with known travel history (11 direct + 2 from media reports), the country of origin is consistent with the sampling locations of the same or ancestral nodes in 9 cases, including the 3 cases when it is uniquely identified. In one case (Supplementary Fig. 4b), the travel direction (Egypt) is inconsistent with the phylogenetic position of the sample, and in the remaining three cases, there is not enough phylogeographic data to make a call. For the same 9 out of the 13 patients, we were able to correctly and uniquely identify the source continent (Europe in all cases).

Partial consistency between direct travel history and phylogenetic position motivated us to attempt to infer the sources of Russian samples phylogeographically. In the absence of travel data, we position the hypothetical source of one transmission lineage (lineage 9, Fig. 5d) as the USA; and of five singletons, as Chile (Supplementary Fig. 2k), England (Supplementary Fig. 2l), France (Supplementary Fig. 2m), and Denmark (Supplementary Fig. 3h,p). For 6 additional singletons, we position the hypothetical source to the continent (Europe in all cases). Finally, we estimate the hypothetical origin of one stem cluster as Sweden (Supplementary Fig. 4c), and of two more stem clusters, as Europe. Importantly, phylogeographic inferences are strongly sensitive to sampling bias, and should be treated with caution.

The individuals importing the virus and seeding the Russian transmission lineages were not a random sample of the population. Very early samples were collected from patients who were on average younger than those sampled later (Fig. 7b). This is consistent with the major role of younger Russians in the import of virus into Russia[36], possibly because they comprised a larger share among the people returning from business trips or holidays.

**Temporal dynamics of SARS-CoV-2 spread in Russia.** Following introduction, the virus has spread throughout Russia. Four out of the 9 identified Russian transmission lineages, and 8 out of the 12 stem clusters, span multiple regions (Figs. 1b, 4c, d, 5d). As Moscow and Saint Petersburg are major transport hubs, together responsible for 77% of the international air traffic in Russia, we

hypothesized that the virus was introduced through these cities, and spread throughout Russia from them. Contrary to this hypothesis, among the Russian stem clusters and singletons, the samples from Moscow or Saint Petersburg do not sit on shorter branches than samples from other regions; in fact, branches leading to them tend to be slightly longer (mean branch length 0.88 vs. 0.37 substitutions, $p = 0.006$, permutation test), probably because of more extensive regional sampling early in the outbreak. Thus, we see no evidence for a preferential direction of transmission within Russia, suggesting that the Russian epidemic has been seeded by near-concurrent introduction into multiple regions.

**Vreden hospital outbreak.** A major transmission cluster corresponded to the nosocomial outbreak at the Vreden Russian Research Institute of Traumatology and Orthopedics in Saint Petersburg (hereafter, the Vreden hospital)[37,38]. According to an internal investigation, the suspected patient zero at the hospital had surgery on March 27. While routine COVID-19 testing at the Vreden hospital began on March 18, the earliest samples that tested positive were collected on April 3. Quarantine was gradually introduced between April 7 and April 9, which involved a complete lockdown of the hospital, isolation of units from each other, and shutdown of the hospital-wide ventilation system. Four hundred and seventy four patients and 270 health care workers remained inside the hospital for the following 35 days.

Our dataset contains SARS-CoV-2 genomes obtained from 52 of the Vreden hospital patients or health care workers. Phylogenetic analysis indicates that these samples form three distinct groups, each defined by its own set of mutations. The largest group, group 1, includes 41 sequences obtained between April 3 and April 22 and represents a distinct Russian transmission lineage (lineage 8, star in Fig. 3). This lineage derives from a very prolific ancestral node which has seeded multiple lineages throughout the world, including five of the Russian transmission lineages, so its origin cannot be positioned phylogeographically. Group 2 contains 7 out of 9 sequences in another clade, which also carries one non-Russian (English) sequence (triangles in Fig. 3). Finally, group 3 includes 4 sequences and represents a clade of its own within another Russian transmission lineage (lineage 1, diamonds in Figs. 2 and 4a). While samples from group 1 came from different units located at different floors of the Vreden hospital, samples from groups 2 and 3 each came from its own unit.

Groups 1 and 2 are phylogenetically remote from group 3, with six mutations separating the most recent common ancestors (MRCAs) of groups 1 and 2 from group 3 (Figs. 2–3). Groups 1 and 2 belong to the B.1.1 lineage defined by three mutations at positions 28881, 28882, and 28883, and are further defined by mutations at positions 26750 and 1191, respectively. By contrast, group 3 belongs to the B.1.5 lineage, and is supported by the mutation at position 20268 which is widespread over the world and appeared early in phylogenetic history, as well as by two additional mutations. This provides strong evidence that group 3 originates from a separate introduction to that of groups 1 and 2.

To understand the spread of the outbreak at the Vreden hospital, we performed a Bayesian phylodynamic analysis using the birth-death skyline model[39] of BEAST2[40]. Given the possibility of multiple introductions, we analyzed the whole Vreden dataset comprising groups 1, 2 and 3; and also its two subsets consisting of groups 1 and 2, and just of group 1. The results are summarized in Figs. 8–9 and Supplementary Tables 4, 5 and 6.

We found that the Bayesian analysis supports at least two distinct introductions of SARS-CoV-2 into the Vreden hospital. This is

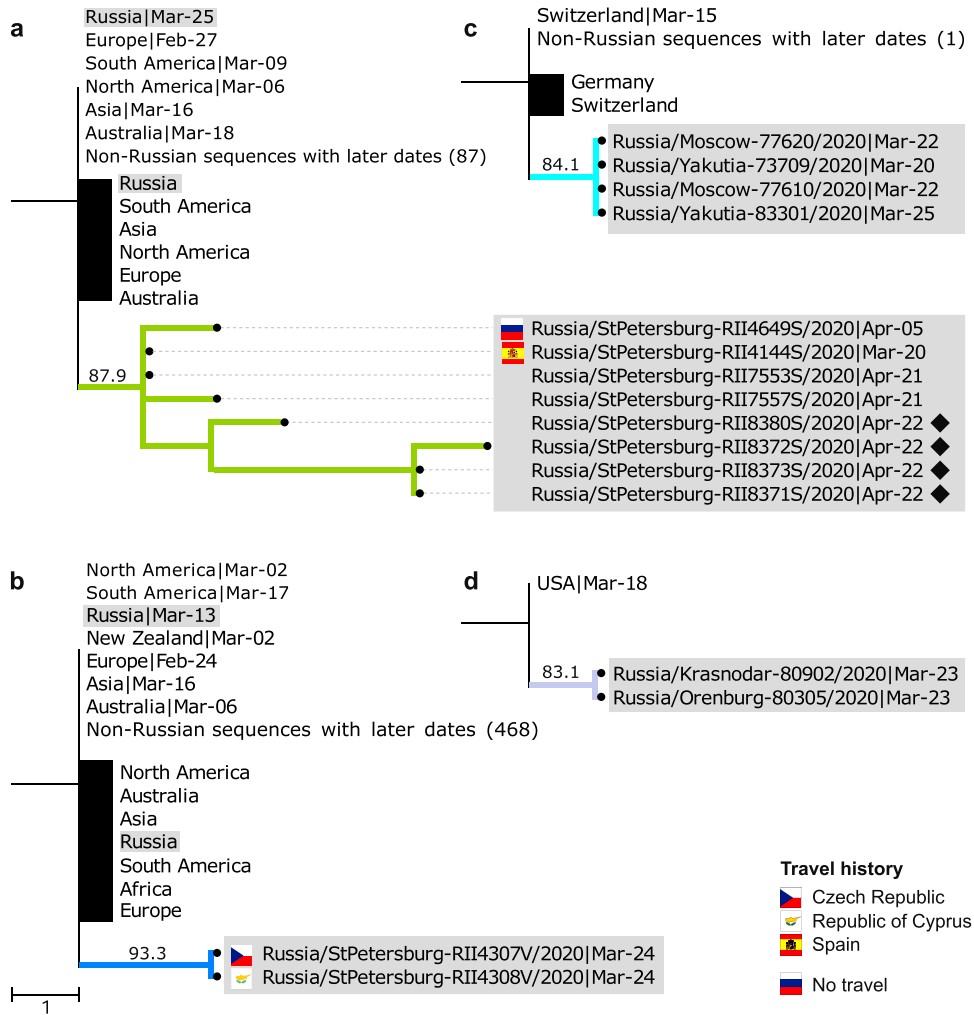

**Fig. 5 Examples of Russian transmission lineages.** Only the phylogeny of the Russian transmission lineage is shown, together with its ancestral phylogenetic node. Russian sequences are marked with dots and highlighted in gray. All other sequences corresponding to an ancestral node (black vertical line) or descendant from it (black rectangle) are truncated, with the region/country and the earliest collection date shown. **a** Lineage 1, a lineage endemic to Saint Petersburg, includes an individual with a history of travel to Spain. **b** Lineage 4 includes individuals with travel history to two different countries, suggesting recurrent introduction. **c** The ancestral node of lineage 3 uniquely maps to Switzerland. **d** The ancestral node of lineage 9 uniquely maps to the USA, and this lineage spans two different regions of Russia. Flags represent individuals with a known history of travel to the corresponding country; the Russian flag shows a known lack of travel history. Diamonds represent samples associated with group 3 of the Vreden hospital outbreak. See Supplementary Fig. 1 for all nine transmission lineages.

based on the deep split between group 3 and groups 1–2. The MRCA of all three groups dates to February 21 (95% CI January 20–March 21). This is more than a month prior to the assumed date of introduction (March 27), implying that group 3 and the remaining Vreden samples were introduced independently.

A third introduction into the Vreden hospital is also highly probable. Indeed, the MRCA of groups 1 and 2 dates to March 24 (95% CI March 6–April 1). As there was no sign of infection at the hospital before the end of March, it is quite likely that these two groups originated through separate introductions. The root of the group 1 dates to March 26 (95% CI March 13–April 2), which is consistent with the suspected illness period of the patient zero. Additional evidence that groups 1 and 2 originate from distinct introductions is provided by the fact that the clade that includes group 2 also carries a non-Russian (English) sequence (Fig. 3).

We estimated the phylodynamic parameters before and after the quarantine measures were introduced. In all three analyzes, the estimates were stable and consistent with each other. Based on the analysis of all three groups, we found that the effective

reproductive number $R_e$ was 3.00 (95% CI 1.85–4.25) before April 8, and dropped to 1.76 (95% CI 0.91–2.71) after April 8 (Fig. 8). The same estimates of the effective reproductive number $R_e$ from the group 1 only are 3.64 (95% CI 2.01–5.43) before quarantine and 1.85 (95% CI 0.77–3.06) after quarantine, respectively. These estimates are consistent with each other, and the potential effects of population structure do not create considerable biases. The substantial decrease of the $R_e$ upon introduction of quarantine can also be seen from the incidence data on moderately-to-severely ill patients (those deemed to require transition to specialized COVID-19 facilities; Supplementary Fig. 5).

## Discussion

The ongoing pandemic of SARS-CoV-2 has involved rapid spread of the virus across the borders of most nations within the few weeks of February and March. While Russia was behind many of the neighboring countries in the initial rise of the case counts, it has rapidly caught up in the following weeks. By analyzing the phylogenetic distribution of 211 early COVID samples from those

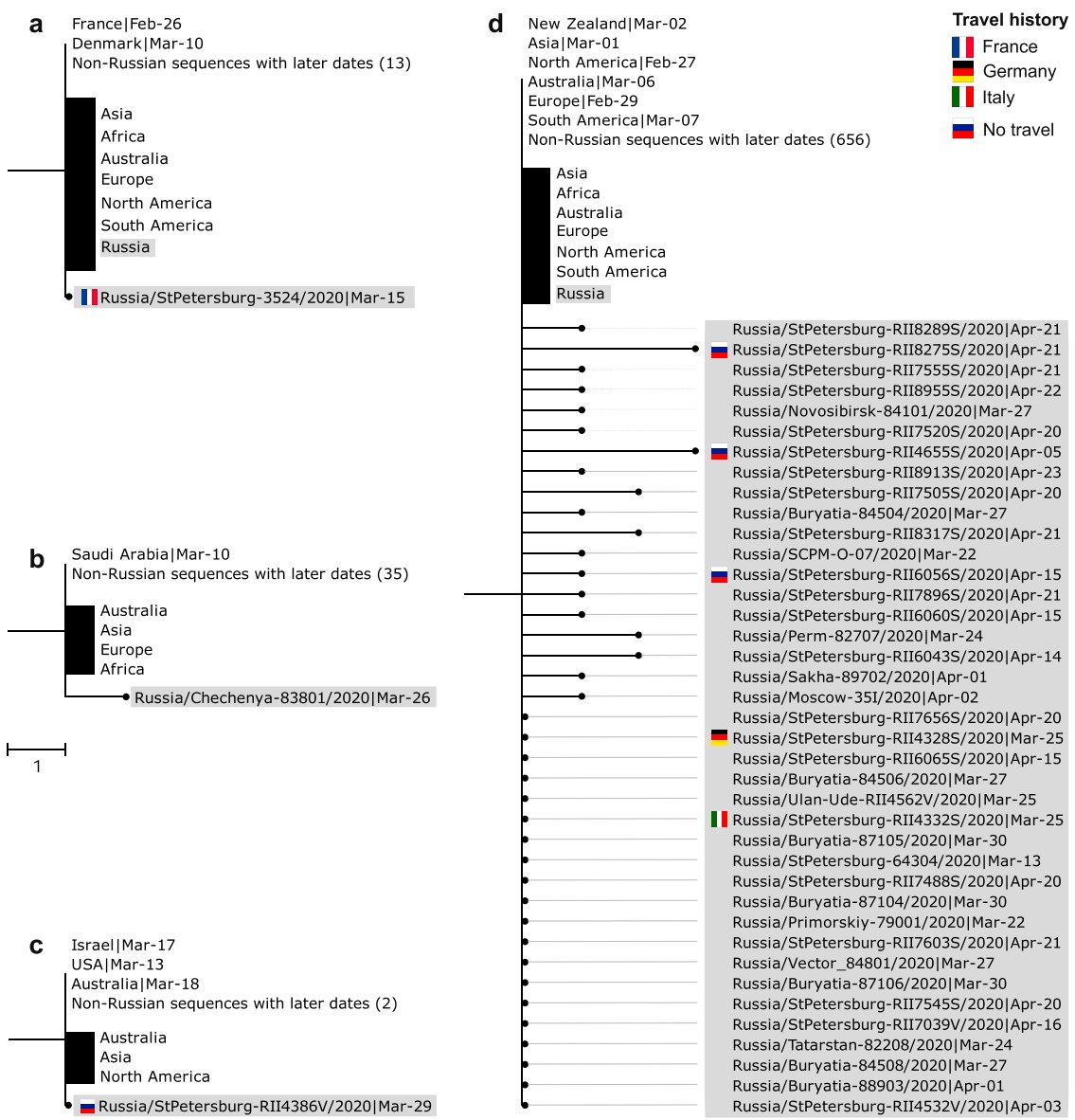

**Fig. 6 Examples of Russian singletons and stem clusters.** Notation is the same as in Fig. 5. **a** The singleton obtained from a patient with known travel history to France has French and Danish sequences at the ancestral node, with French sequences having an earlier collection date. **b** A sIngleton with a uniquely Saudi Arabian ancestral node. **c** A singleton with known absence of travel history. **d** A stem cluster with associated stem-derived singletons where multiple introductions were observed. See Supplementary Figs. 2–4 for all singletons and stem clusters.

dates, we provide details of this process, shedding light of the patterns of transborder transmission of the virus and the factors that affect it.

SARS-CoV-2 accumulates substitutions at the average rate of ~1 per 1000 nucleotides per site per year[41], which means that its genome accumulates on average just one mutation per 2–3 transmissions. Therefore, phylogenetic trees have lower resolution than transmission trees, meaning that transmission history cannot be fully resolved from phylogenetics alone. In particular, in the absence of complete data on travel history, there is no simple rule for counting the number of introductions. A common rule of thumb is counting the number of country-specific clades[42–46]. However, multiple introductions can result in a single clade if the viral diversity abroad is undersampled;[47–49] and a single introduction can result in multiple clades if their last common ancestor has already been introduced[44].

By using direct travel data, we show that both these problems hold. Indeed, we find transmission lineages apparently

co-introduced from multiple countries (Fig. 5b) or singletons without any history of travel (Fig. 6c). The uncertainty in the number of introduction events is the highest for identical sequences with broad geographic distribution, e.g., the last common ancestor of lineage B.1.1. This node constitutes a stem cluster of 100 identical Russia1n sequences, as well as 4323 sequences from outside Russia. It is the immediate ancestor to five Russian transmission lineages and 19 stem-derived Russian singletons, so how many times this sequence has been introduced into Russia strongly affects the overall counts of the number of introductions. Travel data indicate that a stem group can carry a combination of multiple introduced and domestically transmitted sequences, complicating the inference of the number of introductions.

Under a simple statistical model combining genetic and available travel data, we estimate that the sampled diversity of SARS-CoV-2 in Russia originated from 67 introductions. Since this corresponds to roughly one introduction per each three

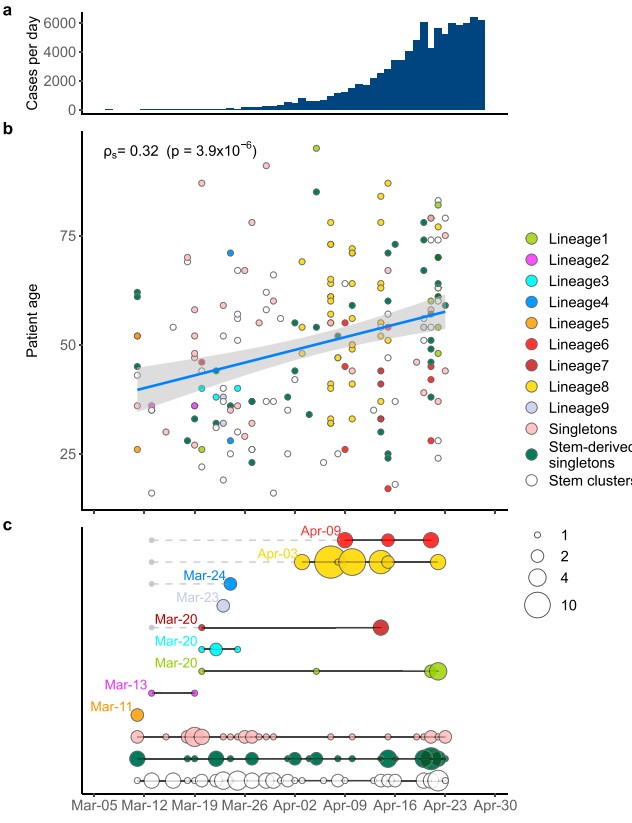

**Fig. 7 The timeline of SARS-CoV-2 introduction into Russia.** Depending on their phylogenetic position, Russian samples are classified as belonging to Russian transmission lineages, singletons or stem clusters (Fig. 4). Circles correspond to Russian samples colored by category. **a** Number of new registered COVID-19 cases per day in Russia between March 5 and April 30. **b** Correlation between sample collection date and patient age. The linear fit ($r = 0.32$; $p = 3.9 \times 10^{-6}$) is shown (blue line), with the 95% confidence interval indicated as a shaded area. Spearman correlation coefficient is shown. **c** Estimated introduction dates for Russian transmission lineages, singletons and stem clusters. The circle size is proportional to the number of samples. Black lines correspond to the full date range. For each Russian transmission lineage, the indicated date corresponds to the collection date of the earliest sample. For stem-derived Russian transmission lineages (lineages 4, 6, 7, and 8), the earliest date of the corresponding stem cluster is also shown with a gray dot.

sequences sampled, the actual number of introductions was probably much higher, and its estimate will likely increase as more sequences are sampled.

Overall, the slow mutation rate of SARS-CoV-2 together with unequal sampling among countries complicates phylogeographic inference of introduction sources[44,50]. We attempted such inference for the Russian samples nevertheless, and found that its results are largely supported by direct travel data when such data is available. The somewhat higher phylogeographic resolution for the Russian lineages compared to that in previous works[44,50] may be due to the fact that most Russian lineges originated late, when the source European lineages were already well established. Still, for most of the introductions, phylogeography is not informative of their origin; moreover, phylogeographic inferences are expected to be biased in the presence of uneven sampling between countries. This illustrates the need of combining multiple data types, including travel history, for understanding viral origin and spread[51].

China is the 5th most popular destination for Russian citizens, accounting for 5.5% of all international travel. Overall,

approximately five million people traveled between the two countries in 2019, with 65% of them traveling by land[52,53]. Although it is hard to ascribe epidemiological results to specific NPIs, our analysis suggests that the border closure with China implemented in February has effectively curbed the virus introduction into Russia from the Asian direction. Indeed, only four of our samples belong to lineages A, B, and B.2 (GISAID clades S, L, and V, respectively), which predominantly originated in Asia; and two of those sequences are nested within other European subclades, indicating that the import was through Europe. This fraction is not representative of global case counts at that time, and is instead reflective of travel patterns and history of border closures. It is also in contrast to the situation in other countries where the outbreaks started earlier, and were probably seeded by direct introduction from Asia[54–56].

For most of the discovered transmission lineages, the earliest sampled sequence was collected between March 11 and 24 (Fig. 7b). In the larger UK dataset, the mean time between the importation date of a lineage and its earliest sampling date within the UK was estimated to be approximately two weeks, although this depends on many factors including lineage size and sampling intensity[44]. If this can be extrapolated to Russian transmission lineages, this implies that these lineages typically originated from imports in the last week of February and the first week of March. A contributing factor could have been intensive travel around the Russian state-mandated long holidays of February 22–24 and March 7–9. Further establishment of Russian transmission lineages could be limited by NPIs, in particular, by introduction of mandatory quarantine for incoming travelers on March 5, as well as by the overall radical reduction in international travel after these dates.

Detailed analysis of localized transmission clusters helps understand viral spread. Well-studied examples include the Diamond Princess cruise ship;[57–60] the Grand Princess cruise ship;[61] an international conference in Boston;[43] a community living facility in the Boston area;[43] and the nosocomial outbreak in the Netcare St. Augustine's Hospital in South Africa[62]. In all but one of these cases, the outbreaks were genetically homogenous, indicating that they each arose from a single case. In the community living facility, multiple introductions have occurred, but there was a dominant clade that included nearly all the samples, while other clades were rare[43]. By contrast, at the Vreden hospital outbreak, we observe multiple (2–3) introductions, each of which gave a prolific clade. This indicates that this outbreak could have originated from multiple superspreading events. Furthermore, we estimate the initial effective reproductive number $R_e$ during the pre-quarantine period at ~3.00, which is rather high. Multiple superspreading events and the high $R_e$ can be due to some of the conditions specific to a hospital not specifically equipped for infection control, including dense contacts (in particular, spread by health care workers), absence of protective measures, and lack of awareness. In the second phase of the outbreak, we observe a significant decrease in $R_e$ down to ~1.76. This change can be explained by two factors. Firstly, it can be due to increased awareness and quarantine measures which were in effect after April 7. Secondly, it can be due to a large number of people already ill, preventing further infection; indeed, around 30% of people at the hospital had been infected by April 22. We cannot quantify the contribution of these factors to the slowing rate of infection spread with available data and methods.

## Methods

**Sample collection and sequencing**. Nasopharyngeal and/or throat swabs were collected in virus transport media. Total RNA was extracted using RiboPrep DNA/RNA extraction kit (AmpliSens, Russia). Extracted RNA was immediately tested for SARS-CoV-2 using LightMix ® SarbecoV E-gene plus EAV control (TIB

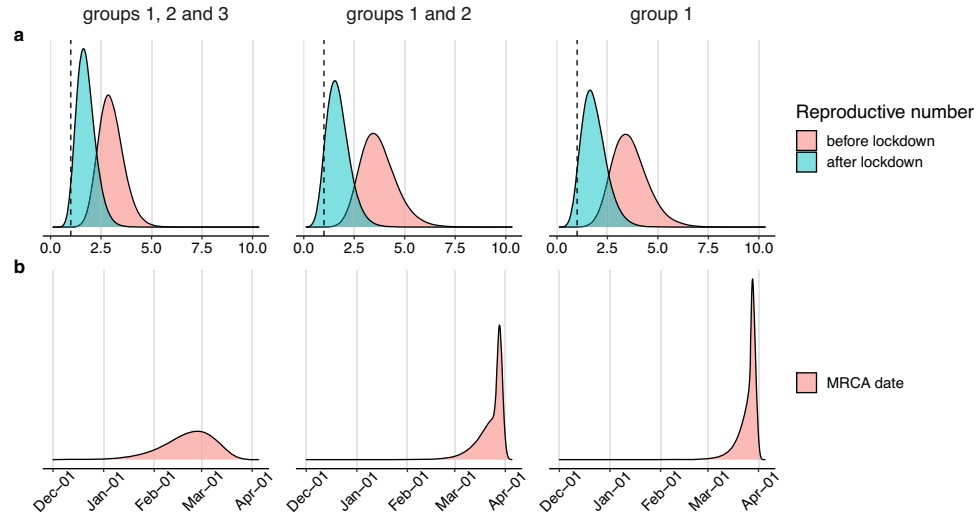

**Fig. 8 Vreden hospital outbreak parameter estimates produced by birth-death skyline model in BEAST2.** Panels show posterior distributions of effective reproductive number $R_e$ (upper panel) with the dashed vertical line corresponding to $R_e = 1$ and the date of the MRCA (lower panel) for analyses based on Vreden hospital samples from groups 1, 2, and 3 (left column), groups 1 and 2 (middle column) and group 1 (right column).

**Fig. 9 Maximum clade credibility tree for the Vreden hospital outbreak.** Groups 1, 2, and 3 are marked by a star, a triangle and a diamond, respectively. Pink bars represent 95% credible intervals. The timeline of the outbreak is shown in gray, with the time interval from patient zero (March 27) till the introduction of quarantine (April 8) highlighted with a darker tone.

Molbiol, Berlin, Germany) provided by the WHO Regional Office for Europe and based on *Charite* protocol[63]. LightMix ® SarbecoV E-gene plus EAV control was used with BioMaster qRT-PCR Kit (Biolabmix, Russia). Briefly, each 20 μl reaction mixture contained 10 μl of 2× buffer, 0.5 μl of LightMix SarbecoV E-gene reagent mix, 4.7 μl of nuclease-free water, 0.8 μl of enzyme, and 4 μl of extracted RNA as the template. RT-PCR was performed on a LightCycler 96 RT-PCR system (Roche). The thermal cycling conditions were 55 °C for 15 min, 95 °C for 5 min, followed by 45 cycles of 95 °C for 5 s, 60 °C for 15 s, and 72 °C for 15 s. Specimens with Ct values less than 30 were selected for whole-genome sequencing.

**Ethics.** Samples used in this study were collected as part of approved ongoing surveillance conducted by the Smorodintsev Research Institute of Influenza. Written informed consent was obtained from all subjects. All samples were de-identified prior to receipt by the study team. The study was presented to the Local Ethics Committee at the Smorodintsev Research Institute of Influenza. The Committee concluded (protocol #151) that the study does not make use of new identifiable biological samples and does not bring forward any new sensitive data. Therefore, according to the rules of the Committee and national regulations this project does not require ethical approval.

**Virus isolation.** For some of the SARS-CoV-2 PCR-positive samples (see Supplementary Data 1), viruses were isolated in Vero cell culture (ATCC #CCL-81). Cells were propagated in MEM (Gibco) supplemented with GlutaMax (Gibco), Sodium Pyruvate (Gibco) and 10% FBS (Gibco #10500). 2 days before inoculation cells were seeded in 5.5 cm$^2$ cell culture tubes (Nunc) at 1:4 ratio and 5% FBS. Samples were diluted 1:10 with serum free media containing antibiotic-antimycotic (Gibco) and inoculated to cells in a volume 0.5 ml/tube. After incubation for 2 h at 37 °C, inoculum was removed and 3 ml of serum free media with anti-anti was added to tubes. Viruses were harvested 4–6 days post inoculation (p.i.) when cytopathic effect (CPE) was near 80–100%, while first signs of CPE were typically observed 2–4 days p.i. For subsequent work, 0.15 ml of virus suspension was lysed in 0.5 ml RLT buffer (QIAGEN) and stored at −20 °C until RNA extraction.

**Whole-genome sequencing.** RNA from primary clinical specimens and virus isolates was re-extracted using QIAamp Viral RNA Mini Kit or RNeasy Mini Kit (QIAGEN). Whole-genome amplification of SARS-CoV-2 virus genome was performed using ARTIC Network protocol[64] with modifications. ARTIC Network primer sets were modified by adding ONT universal tags: 5′-TTTCTGTTGGTGC TGATATTGC-3′ and 5′-ACTTGCCTGTCGCTCTATCTTC-3′ for forward and reverse primers, respectively (see Supplementary Data 1 for details). 1D Ligation sequencing kit (SQK-LSK109) with PCR barcoding expansion (EXP-PBC096) was utilized for sequencing library preparation. MinION (Oxford Nanopore) (flow cell R9.4.1) was used for whole-genome sequencing.

**Genome assembly and consensus correction.** Fast5 files produced by minION were basecalled using guppy_basecaller v3.6.0[65]. Basecalled reads were processed by Porechop v0.2.4[66] in two steps. First, for each sequencing run, reads were demultiplexed with default settings, with built-in barcode and adapter sequences cleaved from read ends. Second, PCR primers were trimmed from demultiplexed reads with options--end_size 70--no_split. Processed reads corresponding to one sample were combined.

For each sample, we then mapped reads onto the Wuhan-Hu-1 SARS-CoV-2 genome sequence (NCBI ID: MN908947.3) using minimap2 v2.17[67] with default settings and filtered out chimeric reads and reads that had secondary alignments. SAMtools-mpileup v1.10[68] was used to produce draft consensus sequences which were then corrected as follows. Mappings were converted into.tsv files using sam2tsv[69], and for each position in the genome, we computed the frequencies of all variants present. We further considered positions with coverage 15 or higher and alternative (compared to Wuhan-Hu-1) variant frequency 50% or higher. We corrected the draft consensus sequences based on the defined set of alternative variants. Each introduced correction was assessed by visually analyzing the corresponding region of mapped reads in IGV v2.8.0[70]. In addition, we manually assessed all alternative variants that had coverage 100 and below. We observed several spurious mutations that were not included in final consensus sequences, including the homoplasic mutation G11083T residing at the end of the poly-T tract in the genome that was observed in five of our samples.

**SARS-CoV-2 dataset preparation and filtering.** All complete high coverage genomes of SARS-CoV-2 for all regions were downloaded from GISAID on May 26, 2020, for a total of 20,469 global sequences and 78 Russian sequences (Supplementary Data 2). To this dataset we added the 136 sequences obtained in this study. Sequences shorter than 29,000 bp, sequences with more than 300 positions with missing data (Ns), sequences excluded by Nextstrain, and samples corresponding to resequencing of the same patients were removed. This led to exclusion of one Russian sample sequenced in this study (hCoV-19/Russia/Ulan-Ude-RII4560S/2020), as well as 834 non-Russian sequences (Supplementary Data 3).

The obtained sequences were aligned with MAFFT v7.453[71] with the following parameters: '--addfragments--keeplength'. As the reference sequence, we utilized Wuhan-Hu-1/2019 (NCBI ID: MN908947.3). To remove low-quality bases from

the alignment, 100 nucleotides from the beginning and from the end were trimmed. The final alignment was used to construct the phylogenetic tree with IQ-Tree v1.6.12[72] with GTR substitution model and '-fast' option. We used TreeTime v0.7.5[73] to reconstruct the sequences of the internal tree nodes. Sequences separated from the tree root by more than ten nucleotide mutations were excluded as probable results of incorrect base calling; this included two sequences from Russia (Russia/SCPM-O-02/2020 and Russia/SCPM-O-05/2020) (Supplementary Data 3). The final dataset contained 19834 virus SARS-CoV-2 sequences. The resulting tree is available as[74].

**Phylogenetic analysis.** We categorized each Russian sequence into one of the five phylogenetic categories based on its phylogenetic position, defined as follows. A Russian transmission lineage is a set of two or more sequences that form a Russian-only clade. A Russian singleton is a single Russian sequence that differs from all other Russian sequences and is not a part of a Russian transmission lineage. A Russian stem cluster is a set of Russian sequences identical to each other and to some non-Russian sequences. A Russian stem-derived transmission lineage is a Russian transmission lineage whose immediate ancestor is a Russian stem cluster. A Russian stem-derived singleton is a Russian singleton whose immediate ancestor is a Russian stem cluster. These categories are schematically represented in Fig. 4.

As sampling dates, we used the collection dates reported in GISAID. For some samples, either the day or both the day and the month of collection were missing. In such cases, the date was set to the latest date possible, e.g., "2020-03" to "2020-03-31" and "2020" to "2020-12-31". The date for the lineage introduction was estimated as the earliest collection date among all samples in the particular lineage, which in fact reflects the latest possible date the lineage could have been introduced.

In phylogeographic analysis, we assumed that the possible source(-s) of introduction for Russian transmission lineages and Russian singletons were the sampling country(-ies) of the non-Russian sequences ancestral to the considered lineage; for Russian stem-derived transmission lineages and Russian stem-derived singletons, these were the non-Russian sequences identical to the ancestral stem cluster. For stem clusters, the possible source of introduction were the countries of origin for the sequences identical to those in the stem cluster. As a possible source of introduction, we only considered those countries with the earliest collection date earlier than the earliest collection date among all samples in the lineage (or than the collection date of the singleton). For patients with known travel history, we considered the country (continent) of origin as uniquely identified by phylogeography if it was either the only country (continent) on the ancestral stem, or the one with the earliest collection date. For samples with no travel data, the same logic was applied, except we only infer the country or continent of origin if it was the only one on the stem. If there were more than eight countries in the list, countries were merged into regions: Africa, Asia, Europe, North America and South America. To study the possibility of introduction from China, we performed the same analysis, but considered China, Hong Kong and Taiwan separately from the rest of Asia.

To understand whether introductions to Russia occurred through major transportation hubs (Moscow and Saint Petersburg), we considered all Russian samples not included in Russian transmission lineages. For these samples, we calculated the branch lengths from each sample to its immediate ancestor, and labeled all samples by two categories: major hubs (Moscow, Moscow region, Saint Petersburg and Leningrad region) and other locations (samples from all other locations in Russia). On this data, we performed a permutation test, shuffling labels across the dataset 1000 times. The two-sided p-value was calculated.

**Phylodynamics of SARS-CoV-2 in Vreden hospital.** As discussed in the Results, the Vreden samples belong to three distinct phylogenetic groups. To account for this, we constrained the phylogeny as follows: (((group1), group 2), (group 3)). We independently ran BEAST2 v2.6.2 on three datasets: (i) the whole Vreden dataset comprising groups 1, 2 and 3; (ii) groups 1 and 2; and (iii) group 1 only. The model details were as follows. The effective reproductive number was allowed to change on March 27 (which delimits the suspected out-of-hospital period) and again on April 8 (which corresponds to the introduction of quarantine). The prior on the clock rate was set to be a normal distribution with mean $9.41 \times 10^{-4}$ and standard deviation $4.99 \times 10^{-5}$, based on the estimates from the UK study[44]. Other priors are provided in Supplementary Table 7. Supplementary Tables 4, 5 and 6 contain the Bayesian estimates of the model parameters for three datasets comprising groups 1, 2, and 3, groups 1 and 2, and group 1, respectively.

We ran a birth-death skyline model with multiple rho-sampling events. This sampling strategy allows us to take into account that the sampling was not continuous over time, but instead was performed on specific dates (April 3, 7, 10, 14, 22).

The time-dependent $R_e$ was independently estimated from incidence data using EpiEstim package v2.2-3 in R[75] with 7-days sliding window and parametric serial interval distribution with mean 4.6 and standard deviation 2.0.

**Public information and data visualization.** The initial Russian map was downloaded from GADM (sf, level 1)[76]. Numbers of confirmed cases in Russia by region were downloaded on May, 26, 2020 from[77]. Patients age data for Russian samples

were extracted from GISAID metadata; for 12 samples, age data were missing. The Spearman correlation between age and collection date was calculated in R version 3.6.3 with cor.test() function. Maps were visualized with the ggplot2 v3.3.0 package in R. Phylogenetic trees were visualized with the ETE3 toolkit v2.3.2[78] in Python v3.6 and iTOL v4[79]. The maximum clade credibility tree was visualized with FigTree v1.4.4[80].

**Reporting summary**. Further information on research design is available in the Nature Research Reporting Summary linked to this article.

## Data availability

GISAID accession IDs of SARS-CoV-2 consensus sequences produced in this study are provided in Supplementary Data 4. Data that support the findings of this study have been deposited at the SRA under accession numbers SRX8723172-SRX8723344, BioProject PRJNA645970. Data referenced in this study is available in Genbank under accession code MN908947.3.

## Code availability

Custom Python/R scripts used for transmission lineages definition and data visualization are available at https://github.com/garushyants/covid_russia_early.

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

## Acknowledgements

We thank Dr. Josh Quick and Dr. Nick Loman (University of Birmingham) for providing the ARTIC Network primer set version 2 for whole-genome amplification of SARS-CoV-2 viruses; Prof. Rasmus Nielsen for his insights in setting up the BEAST2 analysis; Dr. Carlo Pacioni for his help with setting tip dates sampling for BEAST2 analysis. We further thank colleagues from the Chumakov Federal Scientific Center for Research and Development of Immune and Biological Products of Russian Academy of Sciences (Moscow, Russia) for their valuable help with BSL-3 sample processing. No compensation was received for their roles in the study. This research was supported in part through computational resources of HPC facilities at NRU HSE. N.S., V.Sp., D.G., and V.S. performed the research within the framework of the HSE University Basic Research Program. We thank all of the authors who have contributed genome data on GISAID (see Supplementary Data 2 for the list). This study was funded by RFBR project 20-04-60556.

## Author contributions

A.B.K., A.V.F., M.V.S., A.A.I., D.M.D., and D.L. performed RT-PCR testing, isolated the virus, collected, sequenced, assembled, and curated the viral genomes. K.R.S. and A.V.F. performed base calling. K.R.S. and S.K.G. analyzed genomic data and performed phylogenetic analyses. G.A.B. and S.K.G. analyzed geographic data, travel data and GISAID metadata. O.V.S. collected patient samples at the Vreden hospital and contributed to detailed epidemiological analysis. K.R.S, N.S., V.Sp., D.G., and V.S prepared a phylodynamics pipeline, explored probabilistic models and their limitations for the study. N.S. and V.S. performed birth-deaths skyline analysis of the Vreden hospital outbreak. G.A.B. conceived, designed and supervised the project and drafted the manuscript. K.R.S., S.K.G., V.S., and G.A.B. wrote the manuscript with contributions from all authors. All authors read the manuscript and agree to its contents.

## Competing interests

The authors declare no competing interests.
