## [Peer Review File · Nature Communications]

Reviewer #1 (Remarks to the Author):

This is a timely and well-written genomic epidemiology study of the Russian epidemic. As the Russian epidemic is one of the largest and to the best of my knowledge no genomic epidemiology studies of it has been published, this analysis is important to describe the genetic diversity within the Russian epidemic. The Vreden hospital case study is also a good example of a hospital outbreak with multiple introductions. I like the introduction of new terminology to describe different types of transmission lineages and singletons (stem-derived lineages, stem cluster etc.) and I think this terminology will be useful to describe other outbreaks as well. Figure S2 is good and may work better in the main text to explain the terminology. While the analyses and discussion are generally sound there are a few aspects that need some more attention. In particular, the discussion around the number, sources and timing of introductions needs to be a bit more nuanced and acknowledge sampling biases better. There are also a few issues with the Vreden outbreak analysis that I detail below.

Major comments

1. I think a lineage should be classified as a stem-derived lineage and a singleton as a stem-derived singleton whenever the immediate ancestor is a stem cluster, regardless of the collection times. Because the virus mutates slowly and because genomes can be collected from patients over such a long time period the order of sequence collection does not necessarily follow the order of transmission.
2. Transmission lineages could easily originate from more than one introduction if the source location is undersampled. Similarly, the statistical procedure used to estimate the number of introductions for stem clusters, stem-derived lineages and stem-derived singletons is conservative and can also be affected by sampling biases. Although this is acknowledged, it should be emphasized that the estimates are a lower limit by for example writing "originated from \geq at least five distinct introduction events" on page 7, line 17.
3. I don't agree that the position of a lineage in the tree is predictive of its origin location (page 26, line 10). This will hold for a few lineages from well-sampled locations, but is not the rule. In fact, there is a counter-example in the data with the Russian sequence with travel history to Egypt. Because there are only 2 Egyptian sequences in the dataset, this Russian sequence does not cluster with either of the Egyptian sequences and the travel direction appears to be inconsistent with its position in the tree. Thus, without travel data only the probable source of transmission lineages can be established and this will be affected by sampling biases. This uncertainty should be emphasized, for instance on page 10, line 16. Because of the slow mutation rate it is also not a prerequisite that sampling dates of sequences in the source location predate those in the Russian lineage.
4. Lineage introduction dates are taken to be the collection date of the earliest sequence in a lineage. In reality lineages would have been introduced earlier and the earliest collection date represents the latest possible date a lineage could have been introduced. I think this isn't stated clearly enough in the paper.
5. Why was the decision made to use the date of symptom onset as the sampling date of sequences instead of the date of sample collection? Doing so means assuming that patients cannot transmit the virus after first developing symptoms and that the virus stops replicating (and mutating) after patients develop symptoms. The first would only hold if patients were placed in perfect isolation upon developing symptoms. Given that the date of symptom onset is only known for a fraction of patients suggests that this was not the case. The result of this assumption would be to push branching times (infection events) further into the past which would overestimate R

before the lockdown and underestimate R after the lockdown. The second assumption would lead to overestimating the clock rate (since the treelength is too small), but given the strong clock rate prior used here the effect would be minimal.

Because of the above biases and because there usually isn't a lot of information in the data for accurately estimating tip dates (estimates are usually sampled more or less uniformly across the tip date prior), I would like to see the analysis of the Vreden outbreak repeated while fixing sampling dates to the sample collection dates. Because sequences were collected on only 5 dates the birth-death skyline model as used here with a sampling rate through time may not work very well. Instead, the authors could try using a model with 5 scheduled sampling time, estimating the sampling probability at each sampling time (often called rho-sampling). If the decision is made to continue with the tip-date sampling model, the uncertainty in tip dates need to be shown in Fig. 7. It would also help to show the times when the reproductive number and sampling proportion were allowed to change on Fig 7.

6. Because there aren't a lot of mutations in each group of sequences from the Vreden outbreak, most of the information for the birth-death skyline model comes from the sampling dates. Is it possible to get the time-series of infected (and perhaps recovered) patients for the hospital outbreak? This could be used to check the sampling proportion estimates. Because the total population size is known it could also be used to estimate the reproductive number using a package like EpiEstim. Having an extra analysis based on the time-series of cases would add extra confidence to the birth-death skyline results.

7. Using only the genetic data it does not follow that the Vreden hospital outbreak contributed to COVID-19 spread in St. Petersburg. None of the 5 non-Vreden sequences in groups 1 and 2 are derived from any of the hospital sequences. They are either identical to the sequence at the root of the hospital outbreak groups, or contain unique mutations not seen in any of the hospital sequences. This makes it impossible to claim that the transmission chains leading to these 5 cases originated within the hospital (without additional information). It is just as likely that they were infected by the common ancestor of the hospital outbreak groups (unless groups 1 and 2 originated from only one introduction to the hospital, but this seems unlikely). I note that this may not be the case for the 2 cases linked to hospital employees, although I can't tell from the text how direct the link is.

Minor comments

8. How do the numbers of travellers entering Russia by air compare to overland travel? If there is a substantial amount of overland arrivals it is not a foregone conclusion that Moscow and St. Petersburg would receive most importations and the results may only be due to much more intensive sequencing of samples in these two cities than the rest of the country (this seems to be the case with St. Petersburg, which receives less travellers and has less positive cases than Moscow, but has about 4 times the sequences available).

9. How much travel is there usually from China to Russia? If the usual amount of travel is minimal, then border closures would not have had a large impact.

10. Assigning lineages to locations is highly dependent on the sampling period in question. E.g. pangolin lineage B.2 started out predominantly Asian, but has since shifted to being predominantly European (https://cov-lineages.org/lineages/lineage_B.2.html). The same can be said of lineage A (now predominantly North American) and B (now predominantly European).

11. The only estimate of uncertainty in the number of introductions is given for treating each sequence in a stem cluster as an independent introduction. What is the number when also treating each stem-derived singleton as an independent introduction?

12. Fig 2 could be improved. It would be easier to interpret if large triangles are used to indicate whenever a clade has been collapsed. I also don't think it's necessary to label all of the sequences on the figure.

13. Fig S1 is very unwieldy. It would be better to just provide the Newick or Nexus tree as a supplementary data file.

14. In Figures 3, 4, S3 and S4 could you please highlight all Russian sequences, not just the ones within the lineage or the singleton?

15. Page 20, line 9: What are the units of the branch lengths?

16. Because it seems relatively unambiguous that there were at least 3 independent introductions to the Vreden hospital outbreak an analysis that allows for this population structure may be more appropriate. The birth-death skyline model used in the paper assumes a well-mixed population, which is clearly violated in the analyses on more than one group (although this doesn't seem to affect parameter estimates). One possibility is to run a birth-death skyline analysis across 3 trees (one per group) with some shared parameters. Another option is to use a multi-type birth-death process (BDMM in BEAST2) that can model multiple introductions to the hospital. A third option would be using the PhyDyn package to fit an SEIR model with two locations.

17. Zero patient -> patient zero

18. There are some issues with the references. Several author names are not parsed correctly and for some publications the authors are clearly wrong, e.g. Kristian Andersen is not an author of reference 47. For some other references the URLs don't work, e.g. reference 35.

Louis du Plessis

Reviewer #2 (Remarks to the Author):

Komissarov et al performed genome sequencing of SARS-CoV-2 samples collected in Russian during early phase of the outbreak (March-April 2020), and analysed for the virus molecular epidemiology using phylogenetic and phylodynamics method. Their analysis suggested multiple virus introduction into Russia, and majority came from non-Chinese countries. Some introductions have led to local transmissions. No sign of export from Russia are observed. They also did a focused study of outbreak in Vreden hospital, and found that lockdown of the hospital effectively reduce the virus transmission (in terms of R_e). The entire study is well-done, and represent the study of the largest set of Russian SARS-CoV-2 genomes available currently (there is less than 10 sequences released from Russia after April). I agree with most of the data interpretation, but some which are given in my points below:

Page 2 Line 5: Since even for some monophyletic lineages, introductions could not be completely rule out due to potential missing data. It is recommended to conclude 'has originated from "at least" 67 closely timed introductions...'

Page 10, Line 1-3. "No non-Russian sequences were nested within predominantly Russian 1 clades (Figs. 2, 3, Supplementary Fig. 1). Therefore, we observe no sign of export of SARS-CoV-2 outside of Russia." This interpretation is doubtful. In line 13-15, a clade (lineage) is defined as the monophyletic group carrying solely Russian sequences. And because of such definition of clade, one will never find any non-Russian sequences within the Russian clades. In fact, if you look at Figure 2, for example around lineage (clade) 1, there are many other Russian sequences, such as

RII6061S and RII4678S but also many non-Russian sequences indicated by "(875)". They are all not counted into part of the so-called Russian clade/lineage. I think they could all be grouped into a clade (starting from the their MRCA node – B.1.5|G), and it reflects the possibility that the 875 non-Russian sequences could source from this Russian lineage. Note, there are many Russian sequences (RII7594S, RII8890S, RII8944S, RII7581S etc) at the MRCA node of this lineage, equally suggest they may be the source for the 857 non-Russian sequences. I think the current data cannot rule out the possibility of export from Russia.

The phylodynamic analysis of Vreden hospital outbreak indicated that R_e dropped from 3.72 (before April 8) to 1.38 (after April 8) (Page 22, Line 22-27). This seems interestingly show the effect of lock down. It would be nice if the incidence data of the hospital outbreak could be shown, the epi-curve could help to corroborate the dropped R_e , particularly because phylo/sequence-based method is highly subjected to the sequence availability. The low sequencing proportion after April 8 is apparent in Fig 6.

I notice that the topological supports are not presented in both ML and Bayesian phylogenetic trees.

In addition to sequence data, could the authors present the epidemiological (case data) data in Russia and do more comparative analysis of what they found from phylogenetic and what is seen in the epi curve of Russia?

Suppl Table 3, please indicate that these counts refer to the Vreden hospital outbreak sequence data used for the study.

Could the authors made some comments on the D614G variants in Russia?

Reviewer #1 (Remarks to the Author):

This is a timely and well-written genomic epidemiology study of the Russian epidemic. As the Russian epidemic is one of the largest and to the best of my knowledge no genomic epidemiology studies of it has been published, this analysis is important to describe the genetic diversity within the Russian epidemic. The Vreden hospital case study is also a good example of a hospital outbreak with multiple introductions. I like the introduction of new terminology to describe different types of transmission lineages and singletons (stem-derived lineages, stem cluster etc.) and I think this terminology will be useful to describe other outbreaks as well. Figure S2 is good and may work better in the main text to explain the terminology. While the analyses and discussion are generally sound there are a few aspects that need some more attention. In particular, the discussion around the number, sources and timing of introductions needs to be a bit more nuanced and acknowledge sampling biases better. There are also a few issues with the Vreden outbreak analysis that I detail below.

We thank the Reviewer for favorable assessment of our work. We now move Figure S2 to the main text as suggested. We also introduce the suggested changes in the discussion of introductions and the Vreden outbreak analysis (see below).

Major comments

1. I think a lineage should be classified as a stem-derived lineage and a singleton as a stem-derived singleton whenever the immediate ancestor is a stem cluster, regardless of the collection times. Because the virus mutates slowly and because genomes can be collected from patients over such a long time period the order of sequence collection does not necessarily follow the order of transmission.

We agree: in our data as well as in other datasets, we often see mismatches between the sampling dates and phylogenetic positions. As suggested by the Reviewer, we now no longer consider dates when classifying Russian sequences by phylogenetic position. Due to this change, two Russian transmission lineages, lineages 1 and 5, were reclassified as Russian stem-derived transmission lineages. (Singletons were in fact already treated independently of the collection times as suggested.) This change in definition has not altered the estimated number of introduction events. We have introduced corresponding changes in Results, Methods, Supplementary Note, and Supplementary Table 1. We have also clarified our definition of singletons.

2. Transmission lineages could easily originate from more than one introduction if the source location is undersampled. Similarly, the statistical procedure used to estimate the number of introductions for stem clusters, stem-derived lineages and stem-derived singletons is conservative and can also be affected by sampling biases. Although this is acknowledged, it should be emphasized that the estimates are a lower limit by for example writing "originated from *at least* five distinct introduction events" on page 7, line 17.

Revised as suggested.

3. I don't agree that the position of a lineage in the tree is predictive of its origin location (page 26, line 10). This will hold for a few lineages from well-sampled locations, but is not the rule. In fact, there is a counter-example in the data with the Russian sequence with travel history to Egypt. Because there are only 2 Egyptian sequences in the dataset, this Russian sequence does not cluster with either of the Egyptian sequences and the travel direction appears to be inconsistent with its position in the tree. Thus, without travel data only the probable source of transmission lineages can be established and this will be affected by sampling biases. This uncertainty should be emphasized, for instance on page 10, line 16. Because of the slow mutation rate it is also not a prerequisite that sampling dates of sequences in the source location predate those in the Russian lineage.

We certainly agree that such inference is far from reliable, and is strongly affected by sampling bias. We now revise the corresponding sections of the Results and Discussion sections, clarifying this and emphasizing uncertainty more.

4. Lineage introduction dates are taken to be the collection date of the earliest sequence in a lineage. In reality lineages would have been introduced earlier and the earliest collection date represents the latest possible date a lineage could have been introduced. I think this isn't stated clearly enough in the paper.

We now state this explicitly (gratefully borrowing the Reviewer's phrase).

5. Why was the decision made to use the date of symptom onset as the sampling date of sequences instead of the date of sample collection? Doing so means assuming that patients cannot transmit the virus after first developing symptoms and that the virus stops replicating (and mutating) after patients develop symptoms. The first would only hold if patients were placed in perfect isolation upon developing symptoms. Given that the date of symptom onset is only known for a fraction of patients suggests that this was not the case. The result of this assumption would be to push branching times (infection events) further into the past which would overestimate R before the lockdown and underestimate R after the lockdown. The second assumption would lead to overestimating the clock rate (since the treelength is too small), but given the strong clock rate prior used here the effect would be minimal.

Because of the above biases and because there usually isn't a lot of information in the data for accurately estimating tip dates (estimates are usually sampled more or less uniformly across the tip date prior), I would like to see the analysis of the Vreden outbreak repeated while fixing sampling dates to the sample collection dates. Because sequences were collected on only 5 dates the birth-death skyline model as used here with a sampling rate through time may not work very well. Instead, the authors could try using a model with 5 scheduled sampling time, estimating the sampling probability at each sampling time (often called rho-sampling). If the decision is made to continue with the tip-date sampling model, the uncertainty in tip dates need

to be shown in Fig. 7. It would also help to show the times when the reproductive number and sampling proportion were allowed to change on Fig 7.

Following this suggestion, we re-verified the collection dates of all the Vreden samples. We implemented the birth-death skyline model with rho-sampling. Overall, the conclusions of the two analyses are the same; in particular, the new analysis supports our previous result on a decrease in R_e after quarantine introduction, although with a slightly lower statistical significance. We now use the new analysis in the main text, keeping the previous approach in the supplementary materials.

6. Because there aren't a lot of mutations in each group of sequences from the Vreden outbreak, most of the information for the birth-death skyline model comes from the sampling dates. Is it possible to get the time-series of infected (and perhaps recovered) patients for the hospital outbreak? This could be used to check the sampling proportion estimates. Because the total population size is known it could also be used to estimate the reproductive number using a package like EpiEstim. Having an extra analysis based on the time-series of cases would add extra confidence to the birth-death skyline results.

We now add results of EpiEstim (Supplementary Fig. 5). They generally support the downward trend in R_e .

7. Using only the genetic data it does not follow that the Vreden hospital outbreak contributed to COVID-19 spread in St. Petersburg. None of the 5 non-Vreden sequences in groups 1 and 2 are derived from any of the hospital sequences. They are either identical to the sequence at the root of the hospital outbreak groups, or contain unique mutations not seen in any of the hospital sequences. This makes it impossible to claim that the transmission chains leading to these 5 cases originated within the hospital (without additional information). It is just as likely that they were infected by the common ancestor of the hospital outbreak groups (unless groups 1 and 2 originated from only one introduction to the hospital, but this seems unlikely). I note that this may not be the case for the 2 cases linked to hospital employees, although I can't tell from the text how direct the link is.

We agree. To err on the side of caution, we no longer claim an escape from Vreden.

Minor comments

8. How do the numbers of travellers entering Russia by air compare to overland travel? If there is a substantial amount of overland arrivals it is not a foregone conclusion that Moscow and St. Petersburg would receive most importations and the results may only be due to much more intensive sequencing of samples in these two cities than the rest of the country (this seems to be the case with St. Petersburg, which receives less travellers and has less positive cases than Moscow, but has about 4 times the sequences available).

Air travel accounts for 54% of all international travel by Russian citizens, while the remaining 45% of border crossings are by car, train, or on foot (according to the 2019 data). The data on non-Russian citizens is similar. It's however hard to establish the region of destination for inbound land travel. The large number of samples from Moscow and St. Petersburg indeed reflects larger sequencing effort in these cities; we now emphasize non-uniform sampling between regions in the Results section.

9. How much travel is there usually from China to Russia? If the usual amount of travel is minimal, then border closures would not have had a large impact.

We now added information about travel between Russia and China to the Discussion section. The Russian-Chinese border is rather busy (approx 5 million travelers per year); it's the 5th most popular destination for Russian citizens.

10. Assigning lineages to locations is highly dependent on the sampling period in question. E.g. pangolin lineage B.2 started out predominantly Asian, but has since shifted to being predominantly European (https://cov-lineages.org/lineages/lineage_B.2.html). The same can be said of lineage A (now predominantly North American) and B (now predominantly European).

We agree; in the Discussion section, we are careful to talk about where these lineages (probably) originated, not where they have then shifted to.

11. The only estimate of uncertainty in the number of introductions is given for treating each sequence in a stem cluster as an independent introduction. What is the number when also treating each stem-derived singleton as an independent introduction?

Then the inferred number of introductions goes up to 143; this higher boundary is now used in the text.

12. Fig 2 could be improved. It would be easier to interpret if large triangles are used to indicate whenever a clade has been collapsed. I also don't think it's necessary to label all of the sequences on the figure.

We now revised Figure 2. We kept the labels as they provide valuable information on the geography and collection dates of the samples. We moved the labels of the collapsed clades towards tree tips to make them easily distinguishable from Russian sequences and lineages; using large triangles for the clades with magnitudes of difference in size doesn't help. Russian lineages are now marked as colored circles.

13. Fig S1 is very unwieldy. It would be better to just provide the Newick or Nexus tree as a supplementary data file.

We now added the Newick tree as a Supplementary Data 4.

14. In Figures 3, 4, S3 and S4 could you please highlight all Russian sequences, not just the ones within the lineage or the singleton?

All russian sequences are now highlighted in Figures 4, 5, and S2-S4.

15. Page 20, line 9: What are the units of the branch lengths?

Substitutions per site; now specified.

16. Because it seems relatively unambiguous that there were at least 3 independent introductions to the Vreden hospital outbreak an analysis that allows for this population structure may be more appropriate. The birth-death skyline model used in the paper assumes a well-mixed population, which is clearly violated in the analyses on more than one group (although this doesn't seem to affect parameter estimates). One possibility is to run a birth-death skyline analysis across 3 trees (one per group) with some shared parameters. Another option is to use a multi-type birth-death process (BDMM in BEAST2) that can model multiple introductions to the hospital. A third option would be using the PhyDyn package to fit an SEIR model with two locations.

We ran the birth-death skyline model with 3 trees as suggested by the Reviewer. We report the results in the supplementary materials, though the estimates are not convincing for groups 2 and 3. We think that this is due to the small sample sizes of these groups; moreover, all these sequences were collected on a single date.

17. Zero patient -> patient zero

Fixed.

18. There are some issues with the references. Several author names are not parsed correctly and for some publications the authors are clearly wrong, e.g. Kristian Andersen is not an author of reference 47. For some other references the URLs don't work, e.g. reference 35.

Now fixed, thank you.

Louis du Plessis

Reviewer #2 (Remarks to the Author):

Komissarov et al performed genome sequencing of SARS-CoV-2 samples collected in Russian during early phase of the outbreak (March-April 2020), and analysed for the virus molecular epidemiology using phylogenetic and phylodynamics method. Their analysis suggested multiple virus introduction into Russia, and majority came from non-Chinese countries. Some introductions have led to local transmissions. No sign of export from Russia are observed. They

also did a focused study of outbreak in Vreden hospital, and found that lockdown of the hospital effectively reduce the virus transmission (in terms of R_e). The entire study is well-done, and represent the study of the largest set of Russian SARS-CoV-2 genomes available currently (there is less than 10 sequences released from Russia after April). I agree with most of the data interpretation, but some which are given in my points below:

Page 2 Line 5: Since even for some monophyletic lineages, introductions could not be completed rule out due to potential missing data. It is recommended to conclude 'has originated from "at least" 67 closely timed introductions...'

Revised as suggested.

Page 10, Line 1-3. "No non-Russian sequences were nested within predominantly Russian 1 clades (Figs. 2, 3, Supplementary Fig. 1). Therefore, we observe no sign of export of SARS-CoV-2 outside of Russia." This interpretation is doubtful. In line 13-15, a clade (lineage) is defined as the monophyletic group carrying solely Russian sequences. And because of such definition of clade, one will never find any non-Russian sequences within the Russian clades. In fact, if you look at Figure 2, for example around lineage (clade) 1, there are many other Russian sequences, such as RII6061S and RII4678S but also many non-Russian sequences indicated by "(875)". They are all not counted into part of the so-called Russian clade/lineage. I think they could all be grouped into a clade (starting from the their MRCA node – B.1.5[G]), and it reflects the possibility that the 875 non-Russian sequences could source from this Russian lineage. Note, there are many Russian sequences (RII7594S, RII8890S, RII8944S, RII7581S etc) at the MRCA node of this lineage, equally suggest they may be the source for the 857 non-Russian sequences.

I think the current data cannot rule out the possibility of export from Russia.

As mentioned in the previous version of the Methods section, we conservatively assumed that the phylogenetic nodes carrying both Russian and non-Russian sequences were positioned outside Russia. We agree that this could lead to underreporting of exports. To be safe, we now exclude all discussion of exports (which was marginal to our analyses) from the manuscript.

The phylodynamic analysis of Vreden hospital outbreak indicated that R_e dropped from 3.72 (before April 8) to 1.38 (after April 8) (Page 22, Line 22-27). This seems interestingly show the effect of lock down. It would be nice if the incidence data of the hospital outbreak could be shown, the epi-curve could help to corroborate the dropped R_e , particularly because phylo/sequence- based method is highly subjected to the sequence availability. The low sequencing proportion after April 8 is apparent in Fig 6.

We now add EpiEstim R_e estimate based on the incidence data in the Vreden hospital (Supplementary Fig. 5).

I notice that the topological supports are not presented in both ML and Bayesian phylogenetic trees.

SH-aLRT support values are provided for the ML tree available as Supplementary Data 4. For the Vreden hospital outbreak, we fix the topology of the large analyzed clades, so topological supports are inapplicable.

In addition to sequence data, could the authors present the epidemiological (case data) data in Russia and do more comparative analysis of what they found from phylogenetic and what is seen in the epi curve of Russia?

We now add the all-Russia epi curve to Fig. 6. The case counts per region as of the date of the last sequence are presented in Fig. 1a. We would rather not compare the case data among regions or perform country-wide phylodynamics analyses, because our sequencing coverage is highly uneven, making the data inappropriate for such analyses.

Suppl Table 3, please indicate that these counts refer to the Vreden hospital outbreak sequence data used for the study.

Revised as suggested.

Could the authors made some comments on the D614G variants in Russia?

The D614G substitution is nearly ubiquitous in Russia, and has been from the start; in our dataset, it is carried by all but five Russian sequences.

REVIEWERS' COMMENTS

Reviewer #1 (Remarks to the Author):

I think the revised manuscript is stronger and more robust. In particular, including an analysis of the Vreden hospital outbreak using the time series of reported cases that returns similar findings to the birth-death skyline model strengthens that analysis and adds confidence to other genomic epidemiology analyses using birth-death models.

I am satisfied that most of my comments have been addressed in the revision. I have a few remaining minor comments that should be straightforward to address.

1. I still find it difficult to get an overview of where the Russian transmission lineages and singletons fall within the global diversity in Figure 2. If the authors could add an overview, perhaps in an inset, I think that would improve the figure.
2. I think the caption for Figure 3 should be expanded so the figure can stand alone without the main text.
3. Page 33, Line 11-13: In the analysis in the main text sequence sampling dates were were fixed to the 5 dates when samples were collected. The analysis where tip-dating was used is only reported in the supplement. It should be made clear that this sentence refers to a different analysis that is only in the supplement.
4. I think the authors should state that the R_e estimates reported on page 21 are those when considering all 3 groups. I think it would be good to also report at least the R_e estimates considering only group 1 in the main text.
5. Page 33, Line 9: There are 5 sampling dates. One is missing from the text.
6. Please add dates for the periods covered by the different R_e estimates to the supplementary tables.
7. We have since updated our analysis of the UK dataset (ref 45), using a larger dataset and updated methods (<https://www.medrxiv.org/content/10.1101/2020.10.23.20218446v1>), with new estimates of the time from lineage importation to sampling the first sequence (which we call detection lag) and an updated clock rate (somewhat slower than the one used here). These changes are unlikely to make a large difference to the results reported here.

Louis du Plessis

REVIEWERS' COMMENTS

Reviewer #1 (Remarks to the Author):

I think the revised manuscript is stronger and more robust. In particular, including an analysis of the Vreden hospital outbreak using the time series of reported cases that returns similar findings to the birth-death skyline model strengthens that analysis and adds confidence to other genomic epidemiology analyses using birth-death models.

I am satisfied that most of my comments have been addressed in the revision. I have a few remaining minor comments that should be straightforward to address.

1. I still find it difficult to get an overview of where the Russian transmission lineages and singletons fall within the global diversity in Figure 2. If the authors could add an overview, perhaps in an inset, I think that would improve the figure.

We now added to Figure 2 an inset showing distribution of Russian samples across major SARS-CoV-2 clades.

2. I think the caption for Figure 3 should be expanded so the figure can stand alone without the main text.

Thank you for noticing this, the caption is now extended.

3. Page 33, Line 11-13: In the analysis in the main text sequence sampling dates were fixed to the 5 dates when samples were collected. The analysis where tip-dating was used is only reported in the supplement. It should be made clear that this sentence refers to a different analysis that is only in the supplement.

The whole paragraph turned out to be an artefact from the previous version. We are very sorry for confusion, and cleaned up this part of the text thoroughly.

4. I think the authors should state that the R_e estimates reported on page 21 are those when considering all 3 groups. I think it would be good to also report at least the R_e estimates considering only group 1 in the main text.

We updated the tables according to this recommendation:

“The same estimates of the effective reproductive number R_e from the group 1 only, are 3.64 (95% CI 2.01-5.43) before quarantine and 1.85 (95% CI 0.77-3.06) after quarantine respectively. These estimates are consistent with each other, and the potential effects of population structure does not create considerable biases.”

5. Page 33, Line 9: There are 5 sampling dates. One is missing from the text.

Yes, thank you, we added the missing date to this part of the text.

6. Please add dates for the periods covered by the different Re estimates to the supplementary tables.

We updated the tables according to this recommendation.

7. We have since updated our analysis of the UK dataset (ref 45), using a larger dataset and updated methods (<https://www.medrxiv.org/content/10.1101/2020.10.23.20218446v1> <<https://www.medrxiv.org/content/10.1101/2020.10.23.20218446v1>>), with new estimates of the time from lineage importation to sampling the first sequence (which we call detection lag) and an updated clock rate (somewhat slower than the one used here). These changes are unlikely to make a large difference to the results reported here.

We are thankful to the reviewer for pointing it out. We agree that this update to the clockrate does not really affect our analysis given the Bayesian framework.

Louis du Plessis